# Laying Performance, Physical, and Internal Egg Quality Criteria of Hens Fed Distillers Dried Grains with Solubles and Exogenous Enzyme Mixture

**DOI:** 10.3390/ani9040150

**Published:** 2019-04-06

**Authors:** Mohamed E. Abd El-Hack, Khalid M. Mahrose, Faten A. M. Attia, Ayman A. Swelum, Ayman E. Taha, Ramadan S. Shewita, El-Sayed O. S. Hussein, Abdullah N. Alowaimer

**Affiliations:** 1Poultry Department, Faculty of Agriculture, Zagazig University, Zagazig 44511, Egypt; dr.mohamed.e.abdalhaq@gmail.com (M.E.A.E.-H.); khn_mahrose@yahoo.com (K.M.M.); 2Animal Production Department, Faculty of Agriculture, Suez Canal University, Ismailia 41522, Egypt; dr.faten.attia@gmail.com; 3Department of Animal Production, College of Food and Agriculture Sciences, King Saud University, P.O. Box 2460, Riyadh 11451, Saudi Arabia; aswelum@ksu.edu.sa (A.A.S.); aowaimer@ksu.edu.sa (A.N.A.); 4Department of Theriogenology, Faculty of Veterinary Medicine, Zagazig University, Zagazig 44511, Egypt; 5Department of Animal Husbandry and Animal Wealth Development, Faculty of Veterinary Medicine, Alexandria University, Rasheed, Edfina 22758, Egypt; Ayman.Taha@alexu.edu.eg; 6Department of Nutrition and Clinical Nutrition, Faculty of Veterinary Medicine, Alexandria University, Rasheed, Edfina 22758, Egypt; Ramadan_nutrition@yahoo.com

**Keywords:** DDGS, exogenous enzymes, productive performance, egg characteristics, yolk fatty acid

## Abstract

**Simple Summary:**

The present study was designed to investigate the simultaneous use of distillers dried grains with solubles (DDGS) and exogenous enzyme mixture (EEM) on layer performance and egg characteristics. Results confirmed that DDGS is an acceptable feed ingredient in layer diets and that the maximum inclusion level of DDGS in the diet should be around 12% for the best performance and egg characteristics.

**Abstract:**

The effects of dietary inclusion of distillers dried grains with solubles in laying hen diets with and without exogenous enzyme mixture (EEM) on performance and egg characteristics were evaluated. One of the main objectives of this study was to examine the effects of distillers dried grains with solubles (DDGS) and enzyme cocktail on egg yolk fatty acids. The study used total of 144 Hisex Brown laying hens in a 4 × 2 factorial arrangement, including four levels of DDGS (0, 6, 12, and 18% of diet) and two levels of enzyme cocktail (0 or 250 mg EEM/kg of diet) through 32–42 weeks of the age. The inclusion of 18% DDGS was associated with the worst (*p* ≤ 0.001) egg production and the lowest daily feed intake. Numerically, hens fed 6% DDGS diet consumed more feed and had the greatest egg production. The best feed conversion ratio (FCR) was recorded in the control, while the worst was recorded in the 18% DDGS group. Compared with EEM-free diets, EEM supplementation improved FCR by about 2.79%, but the difference was not significant. Shell thickness and shell percentage were significantly increased in hens fed 6% DDGS diet compared to other groups. Egg weights in the 6% and 12% DDGS groups were significantly higher than those in the control and 18% DDGS groups. Non-significant increases in shell and albumin percentages were recorded in groups fed EEM-supplemented diets. The interaction effect of DDGS and EEM was significant (*p* ≤ 0.01) for the majority of egg characteristics. As dietary DDGS level increased, yolk color density increased. Elevating DDGS level up to 18% increased yolk cholesterol, total fat, and total unsaturated fatty acids. The effects of EEM supplementation on egg yolk fatty acid composition and the interaction effects between DDGS and EEM were not significant. Considering these results, it could be concluded that DDGS is an acceptable feed ingredient in layer diets and that the maximum inclusion level of DDGS should not exceed 12% of the whole diet.

## 1. Introduction

The poultry industry was and is still suffering from severe challenges because of the increasing prices of soybean meal and yellow corn which are mainly used in formulating poultry diets. Therefore, there is an urgent need for nutritious and affordable alternatives. To reduce costs, the best strategy is to formulate diets using alternative and locally available ingredients such as distillers dried grains with soluble. Distillers dried grains with solubles (DDGS) is a co-product of ethanol-producing plants that use corn and wheat for manufacturing [1,2]. It is a rich source of crude protein, amino acids, crude fat, and minerals [3,4,5], as well as non-starch polysaccharides (NSP) in poultry diets. However, monogastric animals do not efficiently digest feedstuffs that are high in NSP. As a result, the metabolizable energy of DDGS is lower than that of corn (2820 vs. 3420 kcal/kg, respectively) on an as-fed basis [6,7]. The fiber content of DDGS is another common reason for low digestible energy compared with other feed ingredients. The fiber content is not converted to ethanol, so DDGS has about 35% insoluble and 6% soluble dietary fiber [2]. The apparent digestibility of dietary fiber is about 43.7% in a monogastric animal, which negatively reflects on dry matter digestibility [2]. However, supplementing monogastric diets with exogenous enzymes could improve the available energy of DDGS by degrading the fiber content of DDGS and increasing the digestibility of other components [8,9]. The use of a multi-enzyme complex (enzyme cocktail) aims to deal with more than the problems in the feedstuff used; it may contain enzymes to improve the digestibility of NSP, fibers, or other components [3]. Eggs are a staple food in human diet and have a natural balance of essential nutrients. Eggs have a high nutritional value and provide consumers with cheaper animal protein than other foodstuffs (e.g., meat and milk). Measurements of external and internal egg quality can be beneficial to producers since they provide information on the outcome of the egg production process, and reflect the health condition of the flock. Yolk color is of great importance in terms of consumer expectations in different European countries, with varying demands for yolk color between light yellow and deep orange. Yolk color is affected by various components provided in feed ingredients and feed additives [10]. Roberson et al. [11] claim that yolk color density increased linearly with increasing the dietary DDGS up to 15%. Cholesterol content can range from 11 to 15 mg/g of yolk, which constitutes around 5% of total yolk lipids. Besides this, the high dietary fiber content of DDGS diets might affect cholesterol levels in eggs. The present study hypothesized that the simultaneous use of DDGS will provide an alternative feedstuff and decrease amounts of yellow corn and soybean meal in layer diets without adverse effects on productive performance or egg characteristics. Additionally, the use of the exogenous enzyme mixture (EEM) may improve the digestibility of low-digestible nutrients in DDGS. Moreover, studies investigating the effect of DDGS and enzyme cocktail on egg yolk fatty acids are scarce, so this is one of the major goals of the present study.

## 2. Materials and Methods

The present study was conducted at the Poultry Research Farm, Poultry Department, Faculty of Agriculture, Zagazig University, Zagazig, Egypt. All experimental procedures were carried out according to the Local Experimental Animal Care Committee, and the study design was approved by the institutional ethics committee of the Poultry Department. 

### 2.1. Experimental Design, Birds, and Diets

An experiment with a 4 × 2 factorial arrangement was performed to evaluate the effect of four levels of DDGS (0, 6, 12, and 18% of diet) and two levels of EEM (0 or 250 mg EEM/kg diet) on performance and egg quality of laying hens through 32–42 weeks of age. Laying hens underwent an adaptation period during the 31st week of age. A total of 144 Hisex Brown laying hens (31 weeks) were randomly divided into 8 experimental groups of 18 hens each (6 replicates with 3 hens). Birds (similar in body weight, 1650 g) were housed in a layer cage with 40 × 40 × 40 cm dimensions. Eight isocaloric-isonitrogenous diets in mash form were formulated to cover the nutrient requirements of laying hens from 32 to 42 weeks of age according to NRC [7]. The chemical composition of DDGS (USA) was as follows: 89.05% dry matter; 95.38% organic matter; 4.62% ash; 27.23% crude protein; 11.27% ether extract; 7.45% crude fiber; 38.48% nitrogen-free extract. The composition and chemical analysis of the experimental diets are shown in Table 1. The EEM (at 0 and 250 mg/kg of diet; Multivita, Egypt) used in this study was composed of xylanase (*Trichoderma longibrachiatum*, 600 units/g), protease (*Bacillus subtilis*, 8,000 units/g), and amylase (*Bacillus amyloliquefaciens*, 800 units/g).

### 2.2. Management, Data Collection, and Measurements

Birds were fed ad libitum and fresh water was available at all times during the experimental period. Hens were maintained on a 17-7 h light–dark cycle throughout the experiment. A vaccination program was carried out under veterinarian supervision according to age stages. Egg number and feed intake were recorded daily for each replicate. Feed conversion was calculated as g feed/g egg. Egg weight (to the nearest 0.05 g) was recorded daily and individually for each replicate (of 3 hens) then expressed the result per hen. Egg mass was calculated as g egg per hen per day. 

For assessing egg quality criteria, a sample of three eggs was randomly taken from each replicate every other week during the production period (32–42 weeks of age), then data were pooled for statistical analysis. An external evaluation was carried out according to Sun [13]. Internal measurements were performed according to Yalcin et al. [14]. Yolk color was determined using a Minolta Chroma Meter CR-310 (Minolta Corporation, Ramsey, NJ), calibrated with a white calibration plate as described by Roberts et al. [15]. The Chroma Meter measures Hunter *L**, *a**, and *b** values; where *L** measures relative white (100) to black (0) color, *a** measures relative green (-) to red (+) color, and *b** measures relative yellow (+) to blue (-) color. For determination of cholesterol and fatty acid composition, five eggs were randomly selected from each replicate. The egg yolk was separated immediately, and lipids were extracted [12]. 

Four grams of well-mixed egg yolk were weighed into a 100 mL volumetric flask and 25 mL of a chloroform:absolute alcohol (1:1) mixed solvent was slowly added, after which the sample was shaken constantly until the proteins coagulated. An additional 60 to 65 mL of the mixed solvent was added, and the mixture was allowed to stand for 1 h, during which it was shaken every 5 min. Afterward, the sample was brought up to a volume by the mixed solvent, and the mixture was left to stand until clear. The mixture was then filtered, the mixed solvent was evaporated, and egg yolk lipid was obtained. The fatty acid methyl esters were prepared from lipid extracts [16] and quantified by gas-liquid chromatography (GLC). The fatty acid methyl esters were analyzed using a fused silica CP-Sil 88 capillary column with an inside diameter of 50 × 0.25, installed on a Hewlett Packard 5890 GLC “analyzer” with a flame ionization detector. The GLC was temperature programmed to start at 170 °C and to increase by 1 °C/min until reaching 205 °C. The injector and detector temperatures were set at 250 °C and 270 °C, respectively. Hydrogen was used as a carrier gas at a flow rate of 1 mL/min and the split ratio was 1:50. For the cholesterol analysis, the AOAC [12] method was applied with some modifications. A sample of 10 g egg yolk was transferred to a 250 mL flask. The sample was stirred in an ethanol:methanol:isopropanol (90:5:5) solution, in an amount equivalent to 4 mL/g sample, and 1 mL 60% KOH/g sample. The mixture-containing flask was connected to the water-cooled condenser and refluxed for 1 h. After cooling the digest to room temperature, 100 mL of hexane was added, and the mixture was stirred for 10 min, after which 25 mL of deionized water was added, and the mixture stirred for a further 15 min. The layers were then separated, and the hexane layer was collected in an Erlenmeyer flask. An aliquot of 25 mL from the hexane layer was evaporated in a rotary evaporator at 37 °C. The residue was dissolved in 2 mL hexane and 3 μL was injected into a gas-liquid chromatography, Hewlett Packard 5890 GLC with flame ionization detector. A fused silica SE-30 capillary column (25 × 0.25 inside diameter) was installed and the column temperature was set at 260 °C. The injector and detector temperatures were set at 260 °C and 300 °C, respectively. Hydrogen was used as a carrier at a flow rate of 1.5 mL/min, and the split ratio was 1:150.

For the digestibility trial, daily amounts of the tested diets were weighed at the first day of the collection period and offered once per day. By the end of the digestibility trial, feed residues were weighed and subtracted from the total feed offered to obtain feed intake. The period of excreta collection lasted three days. Collected excreta of each bird were dried at 65 °C for 24 hours. Dried excreta were left for few hours to get equilibrium with the atmosphere then ground, mixed, and stored in screw-top glass jars for chemical analysis. The proximate analysis of DDGS, tested diets, and dried excreta were performed according to AOAC [12] for determination of DM (ID 930.15), OM (ID 942.05), CP (ID 954.01), EE (AOAC 945.16), and CF (AOAC 978.10), using five samples for each nutrient. 

### 2.3. Statistical Analysis

Data were statistically analyzed using SPSS^®^ software statistical analysis program [17]. Treatment differences were considered significant at *p* ≤ 0.05. The normal distribution of variables was performed followed by ANOVA. The differences among means were determined using the post-hoc Newman–Keuls test. Statements of statistical significance are based on *p* ≤ 0.05 unless otherwise stated. The statistical unit was the replicate (3 hens).

## 3. Results 

### 3.1. Laying Performance

No statistical differences were found among the control and the experimental groups receiving 6 or 12% DDGS regarding egg production percentage (Table 2). The highest DDGS level (18%) was associated with the worst (*p* ≤ 0.001) egg production Numerically, hens fed a diet including 6% DDGS had the greatest egg production. Neither EEM supplementation nor the combination of DDGS and EEM had a statistical effect on egg production percentage. For daily feed intake, hens fed the intermediate levels of DDGS (6 and 12%) consumed more feed (*p* ≤ 0.0001), consuming 116.54 and 113.59 g/day, respectively, than those fed the control or 18% DDGS diets. However, the lowest (*p* ≤ 0.0001) value of daily feed intake was found with 18% DDGS diet. Supplementing the diet with 250 mg EEM/kg diet was found to reduce (*p* ≤ 0.0001) the daily feed intake. Moreover, highly significant (*p* ≤ 0.0001) effects were recorded as a result of the interaction between DDGS and EEM. The highest value of daily feed intake was detected with the combination of 6% DDGS and 0 mg EEM, while the lowest was found in the group fed 12% DDGS with 250 mg EEM.

As shown in Table 2, the graded inclusion rates of DDGS in layer diets produced a significant (*p* ≤ 0.0001) effect on feed conversion ratio (FCR). It is worth noting that the best FCR (1.62) was found in the control, while the worst (1.93) was found in the 18% DDGS group. EEM supplementation did not significantly affect FCR. Although not significant, supplementation with EEM improved FCR by about 2.79% when compared with EEM-free diets. The combination effect between DDGS and EEM was significant (*p* ≤ 0.05). Supplementing the control diet with EEM gave the best FCR (1.60). Meanwhile, the interaction between 18% DDGS and 0 mg EEM resulted in the worst FCR (2.05) compared to the other combinations.

### 3.2. Egg Physical Characteristics

Excluding egg shape index and yolk index, all egg characteristics studied were significantly (*p* ≤ 0.01) affected by DDGS levels (Table 3). Egg weights in the 6% and 12% DDGS groups were significantly higher than those of the control and the 18% DDGS group. There was no significant change in egg mass among the groups fed 0, 6, and 12% DDGS diets. Shell thickness and shell percentage were significantly (*p* ≤ 0.05 and 0.01, respectively) increased relative to other groups when hens were fed a diet containing 6% DDGS. Hens fed a diet supplemented with 6% DDGS had the lowest (*p* ≤ 0.01) albumin percentage, the highest yolk percentage, and the highest yolk: albumin ratio. Increasing dietary DDGS level led to an increase (*p* ≤ 0.01) in Haugh units relative to the control diet, and the highest value was obtained when hens were fed an 18% DDGS diet. Supplementing the diets with 250 mg EEM/kg diet significantly (*p* ≤ 0.01) reduced values of egg weight, shell thickness, yolk percentage, and yolk:albumin ratio in comparison to EEM-free diets. 

Non-significant increases in shell and albumin percentages were recorded in groups fed EEM-enriched diets. The interaction effect between DDGS and EEM was highly significant (*p* ≤ 0.01) in all egg characteristics except shell thickness and shell percentage (Table 3).

Results presented in Table 4 indicate that dietary DDGS inclusion had significant (*p* ≤ 0.001) effects on L*, a*, and b* color values. As the dietary DDGS level increased, yolk color density increased. Birds fed 18% DDGS diets had the highest a* and the lowest L* values of all groups. Eggs produced from hens fed diets enriched with EEM had more condensed yolk color in comparison with those fed the un-supplemented diets. The interaction effect on the b* color value was significant (*p* ≤ 0.001). Hens fed a diet containing 6% DDGS and supplemented with 250 mg EEM/kg diet had the highest b* value (39.42) compared to the other groups. 

### 3.3. Chemical Composition of Yolk Lipids

Increasing dietary DDGS levels up to 18% led to significant increases in cholesterol, total fat, and total unsaturated fatty acids (USFA) of egg yolk when compared with the control group (Table 5). Hens fed dietary DDGS (0 and 18%) had the lowest egg U/S ratio. In addition, hens fed a diet supplemented with EEM had significantly (*p* ≤ 0.01) higher yolk cholesterol than those fed un-supplemented diets. No significant differences were detected as a response to the combination of DDGS and EEM.

### 3.4. Egg Yolk Fatty Acids

All unsaturated and saturated fatty acids (as a percentage of total yolk fat) were statistically (*p* ≤ 0.001) affected as a result of dietary DDGS inclusion regardless of the percentage of palmitic acid (Table 6). With regard to unsaturated fatty acids, hens fed the control diet had the highest values of egg yolk palmitoleic and oleic acids compared to the other groups. However, linoleic, α-Linolenic, and erucic acids increased (*p* ≤ 0.001) with increasing dietary DDGS levels. For saturated fatty acids, DDGS significantly (*p* ≤ 0.001) decreased egg yolk content of myristic fatty acids, while stearic, behenic, and lignoceric fatty acids did not show a definite trend regarding DDGS concentration. EEM supplementation and the interaction effect of DDGS and EEM did not show a significant effect on egg yolk fatty acid composition (Table 6).

### 3.5. Nutrients Digestion

Data in Table 7 showed that DDGS level significantly (*p* ≤ 0.01) influenced the digestibility coefficients of all nutrients. The digestion coefficients were more preferable in hens fed diets included 8 % DDGS than the control and the other experimental groups. Conversely, hens fed 18 % DDGS had statistical depression in digestibility coefficients of all studied nutrients. EEM supplementation and the interaction between DDGS and EEM had a positive impact (*p* ≤ 0.05) on the digestibility of ether extract and crude fibers compared with EEM free diets (Table 7). 

## 4. Discussion

### 4.1. Laying Performance

No statistical differences in egg production percentage were detected among the control and 6 or 12% DDGS diet groups (Table 3). This evidence that increasing DDGS inclusion up to 12 % does not have any adverse effects on hens production. In line with our findings, Bregendahl and Roberts [18] found that 23-week-old Hyline layers fed 100 g/kg DDGS diet had the same laying rate as that of the control group (0 g DDGS/kg diet), and Cortes–Cuevas et al. [19] also observed non-significant differences in laying performance (egg production, feed intake, and feed conversion) among DDGS levels in Bovans–White hens at 69 and 77 weeks of age. Roberson et al. [11] reported that hens fed diets containing up to 15% DDGS maintained their production rate, while Huang et al. [20] claimed that dietary inclusion of up to 6% DDGS did not have a negative effect on egg production, although the DDGS groups tended to have better feed efficiency. In contrast, Masa’deh [21] reported negative influences on egg production with increasing levels of DDGS from 0 to 250 g/kg for White Leghorn hens, and Ghazalah et al. [22] showed that egg production and feed conversion of laying hens decreased significantly as dietary DDGS inclusion increased. The conflicting responses of laying hens fed corn DDGS diets is possibly due to the great variation in the availability and nutritional composition of different sources of DDGS [23]. In this study, the inclusion of 18% DDGS in a layer diet negatively affected productive performance traits (Table 3). Egg production decreased as the level of DDGS increased during 32–42 weeks of age. The high level of NSP is one of the reasons for the negative impact of the high DDGS inclusion (18%) in laying hen diets as explained by Abd El-Hack et al. [5]. 

Previous studies did not show significant differences in hens’ feed intake and feed conversion ratio as affected by DDGS dietary levels [24]. Contrary to our results, Roberson et al. [11] found that feed intake and conversion was not affected by dietary supplementation of DDGS, and they suggested that DDGS (as an alternative feed ingredient) should be included in the layer diet at a lower level, such as 6%, to avoid the linear reduction of egg production parameters. Romero et al. [25] also found no significant (*p =* 0.09) effect on FCR (1.98 vs. 2.04 g feed/g egg mass) of laying hens fed 200 g/kg DDGS diets compared to those fed control diets.

In the current study, EEM supplementation did not significantly affect egg production or FCR (Table 2). Similarly, Masa’deh [21] found no statistically significant increase or decrease in egg production due to enzyme addition. On the contrary, Świątkiewicz et al. [26] reported that enriching the diet with enzymes significantly (*p* ≤ 0.05) increased egg production in the second phase of the laying cycle. Our results are also aligned with those reported by Bölükbaşi et al. [27], who reported no significant effect of enzymes on FCR of laying hens. On the other hand, Abd El-Hack et al. [28] demonstrated that supplementing a layer diet with an enzyme cocktail (Gallazyme) improved (*p =* 0.017) feed efficiency and decreased (*p =* 0.014) laying rate compared to the group that did not receive enzyme supplementation. In that study, the interaction between DDGS and Gallazyme had a statistically significant effect (*p* ≤ 0.05 or 0.01) on feed efficiency and egg output [28].

### 4.2. Egg Physical Characteristics

The data reported herein revealed that Haugh units increased in eggs produced by DDGS groups in comparison to the control group (Table 3). Sun [13] reported that DDGS may have positive effects on maintaining the physical state of egg albumin, and that albumin increased when DDGS was incorporated into the diet at rates of up to 50%. In the present study, increasing DDGS levels up to 18% also led to a reduction in shell thickness and shell weight. The low shell weight and shell thickness measures that were obtained from hens fed 18% DDGS diets might be due to the reduction in overall egg weight in the DDGS diet groups compared with the control [13]. On the contrary, Elaroussi et al. [29] indicated that small eggs tend to have thicker eggshell than large eggs when the same amount of calcium is deposited. The reduction in shell thickness might be due to the consumption of sulfur from sulfur-rich DDGS, which might interfere with the absorption of dietary calcium from small intestines [22,30], thus reducing eggshell quality. On the other hand, our results disagree with those obtained by Cheon et al. [24] and Mustafa [31], who found that DDGS did not significantly affect egg characteristics.

Most egg quality characteristics were also not significantly affected by EEM supplementation (Table 3). These results are in agreement with those of Ghazalah et al. [22] and Shalash et al. [32], who claimed that the addition of commercial enzyme preparation to hen diets had no statistical effect on the majority of egg quality traits. Abd El-Hack et al. [28] also found that enriching layer diets with 250 mg Gallazyme/kg diet did not affect many egg quality criteria, although it had a positive effect on eggshell percentage. 

Our findings showed that all egg quality criteria (excluding shell thickness and shell percentage) were significantly (*p* ≤ 0.01) affected by the interaction between DDGS and EEM (Table 3). No specific interaction resulted in the best results for all egg quality criteria, but it is observable that elevating DDGS level up to 18%, plus 250 mg EEM/kg diet, gave the best Haugh unit scores (91.27) compared to the other interactions. Similar results were obtained by Abd El-Hack et al. [28], who reported that the combination of 750 g DDGS/kg substituted for soybean meal, with 250 mg Gallazyme/kg diet, resulted in the best Haugh unit scores (90.03) compared to other experimental groups.

Yolk color is considered one of the main attributes of egg quality for consumers, although it has no effect on egg nutritional value. Cutts et al. [33] indicated that the majority of the people surveyed in some European countries (UK, France, Spain, Poland, Germany, and Greece) expressed a preference for egg yolks with the darkest color (color score 14; measuring 8, 10, 12, and 14 on the Roche Yolk Color Scale). Values of *L**, *a**, and *b** express the density of yolk color: the greater the *a** and *b** values, the denser the color. The redness (*a**) measurement was more accurate for detecting differences in yolk color than either *L** or *b** values [34], and in this study, the highest *a** value was associated with the inclusion of 18% DDGS. Hens that did not receive dietary DDGS produced eggs with the highest *L** and the lowest *a** values. In agreement with our data, Cheon et al. [24] found that yolk color density increased with increasing DDGS level up to 20%. Our results were also in accordance with those of Sun [13], Cortes-Cuevas et al. [19], Świątkiewicz and Koreleski [35], and Loar et al. [36] all of whom confirmed that yolk color density increased with increasing dietary DDGS. Contrary to our results, Lumpkins et al. [37] found insignificant effects of DDGS supplemented diets on egg yolk color score. Roberson et al. [11] also stated that a reduction in yellowness (*b**) was detected in eggs laid by hens aged 64 weeks when fed 15% DDGS.

Xanthophylls in DDGS are highly available and are responsible for yolk yellow color. Distillers dried grains with solubles provided 34 mg xanthophyll/kg, more than that from corn [38], even though corn contains over three times the amount of xanthophylls (10.62 mg/kg; NRC, [7]). Similarly, Roberson et al. [11] indicated that dietary DDGS could increase yolk color density. In addition, the removal of egg yolk starch through ethanol fermentation raised the various nutrient contents, including xanthophylls [24]. The xanthophyll content of corn and DDGS were 17 and 30 mg/kg as reported by NRC [7] and Roberson et al. [11], respectively. Distillers dried grains with solubles could be considered a natural feeding material that provides xanthophylls for egg-producing hens [39]. Consequently, it can be a cost-effective means of improving yolk and skin colors that are preferred by consumers and enhance the value of egg in some regions. Xanthophylls include lutein, zeaxanthin, and cryptoxanthin, and represent 0.1, 0.2, and 0.03% of egg yolk content, respectively [40]. Lutein plays an important role in preventing age-related macular degeneration [41]. Poultry species cannot synthesize xanthophylls and depend on dietary sources for color [13,42]. Therefore, DDGS inclusion in the diet would provide high levels of xanthophylls and increase lutein content in egg yolk. Results of yolk color reported in the present study indicated that the xanthophylls found in DDGS were effectively absorbed and utilized by the hens (Table 4). Farmers can save on the cost of adding artificial pigments to dense yolk color by utilizing DDGS dietary supplementation. 

With regard to EEM supplementation, the reported data (Table 4) agree with those of Shalash et al. [32], who observed that yolk color density increased significantly due to enzyme addition to the diets of laying hens. Ghazalah et al. [22] found that hens fed a 75% DDGS diet substituted for soybean meal, with Avizyme supplementation, had a significantly higher egg yolk color score than the other treatment groups. This might be due to the dietary pigmentation released from cell wall contents [43]. On the contrary, Deniz et al. [23] and Jiang et al. [44] reported that yolk color was not significantly influenced by enzyme cocktail supplementation at any inclusion level of corn DDGS. 

### 4.3. Chemical Composition of Yolk Lipids

The results reported in Table 5 show that dietary DDGS levels significantly (*p* ≤ 0.0001) influence egg yolk lipid composition. Previous studies have shown that increasing levels of dietary DDGS results in a diet with higher fat content, results in greater levels of yolk lipids [13], and that dietary fatty acid composition is the most important factor affecting the fatty acid composition of eggs [45]. The level of egg yolk cholesterol depends on its concentration in very low-density lipoprotein (VLDL), not in blood plasma [46], whereas yolk cholesterol was transported by VLDL [47]. A higher content of unsaturated fatty acids in egg yolk could be attributed to the fact that DDGS contains a large amount of unsaturated fatty acids, and about 56% of these are linoleic acid [48]. Hence, a change of yolk fatty acids composition was expected when using dietary DDGS [20]. The latter authors also reported that the fat and cholesterol content in the egg yolk of 50-week-old ducks increased significantly when ducks were fed diets containing 12 and 18% DDGS [20], which was in agreement with our findings here. Huang et al. [20] indicated that 18% dietary DDGS tended to increase the cholesterol content of yolk during the late laying phase. On the other hand, Cheon et al. [24] showed that DDGS did not exert any influence on the crude fat content of egg yolk. 

### 4.4. Egg Yolk Fatty Acids

Hen eggs are considered among the most beneficial foods for human health based on their high content of omega 6 fatty acid (29.8%), as confirmed by Polat et al. [49]. The composition of dietary fatty acids is the most essential factor affecting the fatty acid component of hen eggs and broiler meat [45,50]. The present study showed that different DDGS inclusion rates can influence the fatty acid composition of egg yolk due to the differences in fatty acid composition in DDGS diets (Table 6). Similarly, Sun [13] showed that different DDGS inclusion rates (0, 17, 35, and 50% DDGS) could influence egg yolk fatty acid composition due to the differences in DDGS diets’ fatty acid composition. Sun [13] also found that omega-3 (linolenic acid and EPA) and omega-6 (linoleic acid) fatty acids were influenced by DDGS diets. On the other hand, Cheon et al. [24] concluded that neither saturated nor unsaturated fatty acids found in yolk were affected by dietary DDGS inclusion. 

### 4.5. Nutrients Digestion

The associative influence between the basal diet and DDGS may be the reason for the improved coefficients in hens fed 8% DDGS diet (Table 7). However, increasing the DDGS level up to 18% associated with impaired digestibility. This depression may belong to the increasing dietary crude fiber along with increasing DDGS level. Elevated fiber contents may increase nitrogen endogenous losses and therefore decrease the apparent digestibility of protein [51]. Omar [52] explained that the increase in dietary fiber is always attributed to a depression in the nutrients digestibility. Similar results obtained by Shalash et al. [53] found that cockers fed diet contained 100% DDGS instead of soybean meal declined the digestion coefficient of the extract to 69.3% vs. 82.37 % for those fed 50% DDGS replacement. Ghazalah et al. [22] demonstrated that elevating DDGS level up to 75% instead of soybean meal gave the lowest crude protein and crude protein digestibilities.

The present study confirmed that the addition of EEM to layer diets improved the digestibility of ether extract and crude fibers (Table 7). Multi-enzyme supplementation may improve the nutrients apparent digestibility by lowering the viscosity of digesta and improving nutrients digestion and absorption [54]. Enzymes also increase the solubilization and disruption of feed endosperm cell wall, which positively reflects on nutrients digestibility as documented by Patterson and Aman [55] and Abd El-Hack et al. [56]. On the other hand, Shalash et al. [32] found that digestibility of crude protein, ether extract, crude fibers, and nitrogen-free extract were not statistically impacted by multi-enzyme supplementation in broiler chickens. 

## 5. Conclusions

Based on our results and discussion, we conclude that DDGS should be considered an acceptable feed ingredient in layer diets. The maximum DDGS dietary inclusion level should not exceed 12% in commercial layer diets. The results of the current study provide relevant information for egg producers and nutritionists on the use of fuel-derived corn DDGS. 

## Figures and Tables

**Table 1 animals-09-00150-t001:** Composition and chemical analysis of each diet.

Items	DDGS ^1^ Inclusion Levels (%)			
	0	6	12	18
*Ingredients composition (%)*				
Yellow corn	60.58	59.40	56.00	53.68
Soybean meal	22.00	16.07	13.00	9.00
DDGS	0.00	6.00	12.00	18.00
Corn gluten meal	4.92	6.39	6.48	6.75
Di-calcium phosphate	1.85	1.83	1.75	1.70
Limestone	8.17	8.19	8.23	8.28
Vitamin premix ^2^	0.15	0.15	0.15	0.15
Mineral premix ^3^	0.15	0.15	0.15	0.15
NaCl	0.30	0.30	0.30	0.30
DL-Methionine	0.12	0.11	0.09	0.09
L-Lysine HCl	0.04	0.14	0.19	0.25
Soybean oil	1.72	1.27	1.66	1.65
*Calculated analysis (%)*^4^:				
Crude protein	18.00	17.96	18.06	18.00
ME (MJ/kg diet)	11.9	11.9	11.9	11.9
Calcium	3.64	3.64	3.63	3.64
Nonphytate P	0.45	0.45	0.45	0.45
Crude fiber	2.94	3.06	3.32	3.54
*Determined analysis (%)* ^5^ *on DM basis:*				
Dry matter	94.82	94.47	94.47	94.53
Organic matter	94.55	94.96	95.06	94.82
Crude protein	18.16	18.27	17.95	17.74
Ether extract	5.18	5.53	5.53	5.47
*Total amino acids content (%)*^4^:				
Methionine	0.31	0.34	0.35	0.36
Lysine	0.84	0.84	0.84	0.84
Cysteine	0.25	0.28	0.30	0.33
Valine	0.67	0.75	0.82	0.89
Arginine	1.01	0.91	0.86	0.79
Threonine	0.48	0.55	0.62	0.68
*Digestible amino acids content (%)*^4^:				
Methionine	0.29	0.31	0.32	0.33
Lysine	0.71	0.61	0.55	0.48
Cysteine	0.21	0.23	0.25	0.27
Valine	0.60	0.67	0.42	0.78
Arginine	0.93	0.81	0.75	0.67
Threonine	0.42	0.47	0.51	0.55

^1^ Distillers dried grains with solubles. ^2^ Layer vitamin premix, each 1.5 kg consists of: Vit. A, 12000.000 IU; Vit. D3, 2000.000 ICU; Vit. E 10 g; Vit. K, 328 mg; Vit. B1, 1000 mg; Vit. B2, 5000 mg; Vit. B6, 1500 mg; Vit. B12, 10 mg; Biotin, 50 mg; Pantothenic acid, 10 g; Niacin, 30 g; Folic acid, 1000 mg. ^3^ Layer mineral premix, each 1.5 kg consists of: Mn, 60 g; Zn, 50 g; Cu, 10 g; I, 1000 mg; Co, 1000 mg. ^4^ Calculated according to NRC [7]. ^5^ Analyzed according to AOAC [12].

**Table 2 animals-09-00150-t002:** Laying performance of hens organized by DDGS ^1^ inclusion levels and enzyme supplementation through 32–42 weeks of age.

Items		Egg Production Percentage (%)	Feed Intake (g/day)	Feed Conversion (g feed/g egg)
DDGS level (%)				
0		92.70^a^	105.08^b^	1.62^c^
6		93.77^a^	116.54^a^	1.73^bc^
12		89.73^a^	113.59^a^	1.79^b^
18		83.80^b^	100.28^c^	1.93^a^
EEM ^2^ (mg/kg of diet)				
0		91.20	114.02^a^	1.79
250		88.80	103.73^b^	1.74
DDGS (%)	EEM ^2^ (mg/kg of diet)			
0	0	94.53	110.10^d^	1.64^e^
	250	90.87	100.07^e^	1.60^f^
6	0	93.70	117.35^a^	1.69^d^
	250	93.87	115.74^b^	1.77^c^
12	0	92.47	116.85^ab^	1.78^c^
	250	87.03	110.33^d^	1.81^b^
18	0	84.07	111.77^c^	2.05^a^
	250	83.54	88.79^f^	1.81^b^
SEM		1.12	1.77	0.03
*Probabilities*:				
DDGS		0.001	0.0001	0.0001
Enzyme		0.718	0.0001	0.254
DDGS × enzyme		0.653	0.0001	0.040

^1^ Distillers dried grains with solubles. ^2^ Exogenous enzyme mixture. Means in the same column within each classification bearing different letters are significantly different (*p* ≤ 0.05).

**Table 3 animals-09-00150-t003:** Egg characteristics of laying hens organized by DDGS ^1^ inclusion levels and enzyme supplementation through 32–42 weeks of age.

Items		Egg Weight (g)	Egg Mass (g/day/hen)	Egg Shape Index (%)	Shell Thickness (mm)	Shell (%)	Albumin (%)	Yolk (%)	Yolk: Albumin Ratio	Yolk Index (%)	Haugh Units
DDGS level (%)											
0		67.57^b^	62.64^a^	77.79	0.36^b^	12.46^ab^	65.38^c^	22.16^ab^	0.34^b^	48.84	85.65^b^
6		69.08^a^	64.77^a^	78.53	0.38^a^	12.99^a^	64.02^d^	22.81^a^	0.36^a^	49.87	88.43^a^
12		68.89^a^	61.82^a^	76.86	0.37^ab^	12.22^ab^	66.13^a^	21.65^c^	0.33^b^	50.00	88.66^a^
18		65.55^c^	54.93^b^	77.23	0.34^c^	11.58^b^	65.60^b^	22.82^a^	0.35^a^	50.16	89.97^a^
EEM ^2^ (mg/kg of diet)											
0		68.442^a^	62.42	77.699	0.36^a^	12.19	65.11	22.70^a^	0.35^a^	50.55^a^	88.73
250		67.098^b^	59.58	77.509	0.37^b^	12.44	65.54	22.02^b^	0.34^b^	48.89^b^	87.62
DDGS (%)	EEM (mg/kg of diet)										
0	0	68.49^c^	64.75^b^	77.99^c^	0.35	12.90	64.32^bc^	22.78^bc^	0.36^b^	48.82^c^	85.23^g^
	250	66.65^de^	60.56^c^	77.59^c^	0.38	12.02	66.44^a^	21.54^d^	0.32^e^	48.87^c^	86.06^f^
6	0	71.48^a^	66.98^a^	79.08^ab^	0.38	12.57	64.57^bc^	22.86^b^	0.36^b^	53.42^a^	90.16^c^
	250	66.67^de^	62.58^c^	78.19^b^	0.39	13.42	63.81^c^	22.77^bc^	0.36^b^	46.32^d^	86.69^ef^
12	0	67.71^d^	62.61^c^	74.65^d^	0.37	12.05	66.82^a^	21.13^e^	0.32^e^	51.32^b^	90.87^b^
	250	70.08^b^	60.99^b^	79.08^ab^	0.38	12.39	65.44^b^	22.17^c^	0.34^c^	48.68^c^	86.45^ef^
18	0	66.09^e^	55.56^e^	79.29^a^	0.33	11.22	64.75^bc^	24.03^a^	0.38^a^	48.63^c^	88.67^d^
	250	65.00^f^	54.28^d^	75.17^d^	0.35	11.95	66.45^a^	21.60^d^	0.33^d^	51.69^b^	91.27^a^
SEM		0.39	0.56	0.41	0.00	0.18	0.32	0.19	0.00	0.46	0.54
*Probabilities*:											
DDGS		0.0001	0.001	0.291	0.0001	0.029	0.0001	0.004	0.001	0.434	0.008
Enzyme		0.0001	0.432	0.765	0.0001	0.414	0.400	0.009	0.003	0.012	0.186
DDGS × enzyme		0.0001	0.003	0.001	0.662	0.213	0.0001	0.000	0.000	0.000	0.016

^1^ Distillers dried grains with solubles. ^2^ Exogenous enzyme mixture. Means in the same column within each classification bearing different letters are significantly different (*p* ≤ 0.05).

**Table 4 animals-09-00150-t004:** Yolk color of laying hens organized by DDGS ^1^ inclusion levels and enzyme supplementation Table 32–42 weeks of age.

Items		*L** ^3^	*a** ^4^	*b** ^5^
DDGS level (%)				
0		59.28^a^	8.66^d^	37.67^b^
6		58.52^b^	10.24^c^	38.35^a^
12		58.11^b^	10.76^b^	37.13^c^
18		57.13^c^	11.55^a^	37.88^b^
EEM ^2^ (mg/ kg of diet)				
0		58.49^a^	9.97^b^	36.70^b^
250		58.03^b^	10.63^a^	37.04^a^
DDGS (%)	EEM (mg/ kg of diet)			
0	0	59.65	8.52	37.29^c^
	250	58.92	8.80	38.04^b^
6	0	58.49	9.89	37.27^c^
	250	58.56	10.60	39.42^a^
12	0	58.41	10.35	37.32^c^
	250	57.80	11.17	36.94^cd^
18	0	57.41	11.12	36.27^d^
	250	56.85	11.97	37.13^c^
SEM		0.17	0.21	0.17
*Probabilities*:				
DDGS		0.0001	0.0001	0.0001
Enzyme		0.0140	0.0001	0.0001
DDGS × enzyme		0.376	0.299	0.0001

^1^ Distillers dried grains with solubles. ^2^ Exogenous enzyme mixture. ^3^ Yolk whiteness, ^4^ yolk redness, and ^5^ yolk yellowness. Means in the same column within each classification bearing different letters are significantly different (*p* ≤ 0.05).

**Table 5 animals-09-00150-t005:** The chemical composition of yolk lipids of laying hens organized by DDGS ^1^ inclusion levels and enzyme supplementation through 32–42 weeks of age.

Items		Cholesterol (mg/100 g fat)	Total fat (%)	∑ USFA ^2^ (%)	∑ SFA ^3^ (%)	U/S ^4^
DDGS level (%)						
0		279.83^d^	28.60^b^	60.56^b^	35.41^b^	1.71^c^
6		310.78^c^	29.59^a^	61.58^a^	35.02^c^	1.76^a^
12		318.32^b^	29.78^a^	61.51^a^	35.60^ab^	1.73^b^
18		340.45^a^	29.33^a^	61.53^a^	35.92^a^	1.71^c^
EEM ^5^ (mg/ kg of diet)						
0		310.97^b^	29.32	61.27	35.48	1.73
250		313.72^a^	29.33	61.33	35.49	1.73
DDGS (%)	EEM (mg/ kg of diet)					
0	0	278.95	28.57	60.52	35.34	1.71
	250	280.71	28.62	60.61	35.49	1.71
6	0	309.18	29.55	61.46	35.03	1.76
	250	312.38	29.63	61.70	35.00	1.76
12	0	317.29	29.81	61.54	35.64	1.73
	250	319.34	29.76	61.48	35.56	1.73
18	0	338.45	29.35	61.55	35.91	1.72
	250	342.45	29.31	61.52	35.92	1.71
SEM		3.91	0.11	0.10	0.08	0.00
*Probabilities*:						
DDGS		0.0001	0.0001	0.0001	0.0001	0.0001
Enzyme		0.0001	0.9470	0.7020	0.9060	0.8950
DDGS × enzyme		0.152	0.988	0.905	0.900	0.790

^1^ Distillers dried grains with solubles. ^2^ Total unsaturated fatty acids. ^3^ Total saturated fatty acids. ^4^ Ratio of unsaturated fatty acids to saturated fatty acids. ^5^ Exogenous enzyme mixture. Means in the same column within each classification bearing different letters are significantly different (*p* ≤ 0.05).

**Table 6 animals-09-00150-t006:** Egg yolk fatty acids of laying hens organized by DDGS ^1^ inclusion levels and enzyme supplementation through 32–42 weeks of age.

Items		Fatty Acid Composition as Percent Of Total Yolk Fat										
		Unsaturated fatty Acids						Saturated Fatty Acids				
		Palmitoleic C16:1n-7	Oleic C18:1n-9	Linoleic C18:2n-6	α-Linolenic C18:3n-3	Archidonic C20:4n-6	Erucic C22:1n-9	Palmitic C16:0	Stearic C18:0	Myristic C14:0	Behenic C22:0	Lignoceric C24:0
DDGS level (%)												
0		3.75^a^	43.68^a^	11.55^d^	0.37^c^	0.31^a^	1.67^d^	26.30	9.48^a^	0.36^a^	0.16^d^	0.86^d^
6		3.53^b^	41.08^b^	15.84^c^	0.36^d^	0.31^a^	2.24^c^	26.27	9.26^b^	0.36^a^	0.17^c^	1.14^c^
12		2.85^c^	38.54^c^	18.54^b^	0.44^b^	0.26^c^	2.36^a^	25.94	8.85^c^	0.32^b^	0.18^b^	2.10^a^
18		2.29^d^	37.44^d^	18.84^a^	0.57^a^	0.27^b^	2.29^b^	25.64	9.40^a^	0.28^c^	0.21^a^	1.76^b^
EEM ^2^ (mg/kg of diet)												
0		3.12	40.21	16.22	0.43	0.29	2.14	26.04	9.24	0.33	0.18	1.46
250		3.08	40.17	16.16	0.43	0.29	2.14	26.03	9.26	0.33	0.18	1.47
DDGS (%)	EEM (mg/kg of diet)											
0	0	3.82	43.74	11.61	0.37	0.31	1.67	26.20	9.47	0.36	0.16	0.86
	250	3.67	43.62	11.49	0.37	0.31	1.67	26.40	9.49	0.36	0.16	0.86
6	0	3.53	41.06	15.84	0.36	0.31	2.24	26.28	9.26	0.36	0.17	1.14
	250	3.54	41.10	15.83	0.36	0.31	2.24	26.26	9.25	0.36	0.17	1.14
12	0	2.85	38.58	18.59	0.44	0.26	2.36	26.04	8.81	0.32	0.18	2.10
	250	2.84	38.51	18.49	0.44	0.26	2.36	25.85	8.89	0.32	0.18	2.10
18	0	2.29	37.46	18.85	0.57	0.27	2.29	25.66	9.40	0.28	0.21	1.76
	250	2.29	37.43	18.83	0.57	0.27	2.29	25.62	9.40	0.28	0.21	1.76
SEM		0.11	0.43	0.53	0.02	0.00	0.05	0.10	0.05	0.01	0.00	0.09
*Probabilities*:												
DDGS		0.0001	0.0001	0.0001	0.0001	0.0001	0.0001	0.114	0.0001	0.0001	0.0001	0.0001
Enzyme		0.535	0.605	0.319	0.948	0.725	0.956	0.955	0.619	0.690	0.636	0.775
DDGS × enzyme		0.776	0.920	0.886	0.994	0.998	0.998	0.928	0.860	0.981	0.999	0.986

^1^ Distillers dried grains with solubles. ^2^ Exogenous enzyme mixture. Means in the same column within each classification bearing different letters are significantly different (*p* ≤ 0.05).

**Table 7 animals-09-00150-t007:** Digestibility coefficients of nutrients as affected by DDGS ^1^ inclusion levels and enzyme supplementation through 32–42 weeks of age.

Items		Digestibility Coefficients, on DM Basis (%)					
		Dry Matter	Organic Matter	Crude Protein	Ether Extract	Nitrogen-Free Extract	Crude Fiber
DDGS level (%)							
0		74.41^a^	80.07^a^	68.38^a^	75.32^b^	25.00^b^	80.16^ab^
6		74.93^a^	79.37^a^	70.88^a^	79.91^a^	27.19^a^	83.54^a^
12		74.72^a^	79.52^a^	68.77^a^	75.87^b^	24.07^b^	80.42^b^
18		72.35^b^	75.50^b^	58.24^b^	65.48^c^	18.02^c^	61.58^c^
EEM ^2^ (mg/kg of diet)							
0		73.96	78.33	66.61	72.84^b^	23.39	75.49^b^
250		74.25	78.90	66.53	75.45^a^	23.75	77.35^a^
DDGS (%)	EEM (mg/kg of diet)						
0	0	74.34	80.03	68.34	75.28^b^	25.00	79.42^ab^
	250	74.48	80.11	68.41	75.35^b^	24.99	80.89^ab^
6	0	74.89	79.41	73.06	78.96^a^	27.30	83.01^a^
	250	74.97	79.33	68.70	80.87^a^	27.08	84.06^a^
12	0	74.73	79.68	69.06	76.29^b^	24.06	79.29^ab^
	250	74.72	79.36	68.49	75.46^b^	24.09	81.54^ab^
18	0	71.88	74.19	55.97	60.85^c^	17.20	60.24^c^
	250	72.83	76.80	60.52	70.12	18.83	62.91^c^
SEM		0.26	0.49	1.24	1.29	0.77	1.88
*Probabilities*:							
DDGS		0.000	0.000	0.000	0.000	0.000	0.000
Enzyme		0.401	0.362	0.960	0.024	0.611	0.916
DDGS × enzyme		0.730	0.374	0.263	0.015	0.762	0.032

^1^ Distillers dried grains with solubles. ^2^ Exogenous enzyme mixture. Means in the same column within each classification bearing different letters are significantly different (*p* ≤ 0.05).

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
