# Peer review of "Laying Performance, Physical, and Internal Egg Quality Criteria of Hens Fed Distillers Dried Grains with Solubles and Exogenous Enzyme Mixture"

_animals, 2019, doi:10.3390/ani9040150_

Reviewer 1 Report

The paper has been improved and the authors made some efforts to answers reviewers' comments.

Please check the English writing as some typing errors were noticed.

Regarding the scientific part, I have a concern with diets formulation because all diets are not balanced on a digestible amino-acids basis. Values and ratio of digestible AA are different. This can impair the analysis of the results. How to be sure the differences you observed are related to the DDGS or EEM and not to the nutritional characteristics of the diets? 

Author Response

Thanks for the respected reviewer for his valuable comments which shared in improving the quality of our manuscript. Herein, our reply for the two concerns:

1- Please check the English writing as some typing errors were noticed.

Response: Although the paper was already linguistically revised in a certified office (Editage, certificate is attached), the paper revised again by an English native speaker (American colleague) as a response to the reviewer suggestion. Some corrections were made.

2- Regarding the scientific part, I have a concern with diets formulation because all diets are not balanced on a digestible amino-acids basis. Values and ratio of digestible AA are different. This can impair the analysis of the results. How to be sure the differences you observed are related to the DDGS or EEM and not to the nutritional characteristics of the diets?

Response: The four diets in Table 1 are balanced for their contents of crude protein, metabolizable energy, calcium, non-phytate phosphorous and the most important essential amino acids (Lysine and methionine+ cysteine) according to NRC (1994). The differences among diets for content of other amino acids are very small and not expected to make sense. We thank the reviewer for his notice  and we will consider reducing these differences as much as possible in our ongoing work.

Reviewer 2 Report

The work simple, but it was well planned and described in a manuscript that is very easy to read and follow. In my opinion, it is recommended for publication.

Author Response

Thanks for the reviewer for his positive opinion on our manuscript.

This manuscript is a resubmission of an earlier submission. The following is a list of the peer review reports and author responses from that submission.

Round  1

Reviewer 1 Report

The major objective of this work is how simultaneous use of DDGS and exogenous 69 enzyme mixture affected the performance of laying hens and egg characteristics. The work very simple, but it was well planned and described in a manuscript that is very easy to read and follow. In my opinion, it is recommended for publication after some minor corrections:

Abstract

-Include the reference of the acronym  DDGS.

Introduction and Discussion

- Line 64, 290: What does it mean ,,yolk color increased’’?

Author Response

The major objective of this work is how simultaneous use of DDGS and exogenous 69 enzyme mixture affected the performance of laying hens and egg characteristics. The work very simple, but it was well planned and described in a manuscript that is very easy to read and follow. In my opinion, it is recommended for publication after some minor corrections:

Response: Thanks for the respected reviewer for the positive opinion.

Abstract

-Include the reference of the acronym DDGS.

Response: Done accordingly.

Introduction and Discussion

- Line 64, 290: What does it mean ,,yolk color increased’’?

Response: It means yolk color density, cleared through the text.

Author Response

 Introduction

Please do not separate the introduction into 3 paragraphs. This is a bit academic and it does not make the reading easier.

Response: Done accordingly.   

Information regarding why DDGS are used in poultry nutrition and why this is important to focus on this feedstuff in this study should be better argued.

Response: Done accordingly.

L 52 You mentioned that poultry do not efficiently digest feedstuffs that are rich in NSP and so the use of exogenous enzymes could be beneficial. Please, indicate the fiber content of the DDGS, maybe compare to other feedstuffs. Then, argue why you use a combination of enzymes and not only a xylanase?

Response: Argued in the text.

The state of the art is too poor. General elements are lacking to convince the reader why it is important to study the effect of DDGS and enzymes on laying performances and egg quality. There is also a lack of assumptions about expected results and application prospects (what can this work do for)?

Response: More information added.

L 71 "we believe that our study will address this gap in the literature". Please do not be so ambitious, the proposed study will meet a particular goal, with a given device in particular conditions.

Response: Corrected.

Materials and methods

L 78-80 Information on how you decided the levels of DDGS (0, 6, 12, 18%) and EEM (0 vs. 250 mg/kg) is lacking. This could be better introduced by an improved introduction presenting a bit existing knowledge.

Response: Thanks for the respected reviewer for the valuable comment. Our lab. have already done many experiments on the best levels of DDGS in layer diets either as a substitution for soybean meal or an inclusion in the whole diet. Therefore, selecting these levels (0, 6, 12 and 18%) was based on obtained results. For enzyme levels, 250 mg/kg is the recommended level of the manufacturing company.  

L 81 If there was an adaptation period during the 31st week of age, I guess the random allocation into 8 groups was done at 31st week and not 32 as indicated.

Response: Exactly, there was a week as an adaptation period. 32 corrected to be 31st week.

L 88 and Table 1:

Yellow corn (8.5%); soybean meal (44%), DDGS (24.40%) …: what is indicated into brackets? The protein content of the feedstuff? Why is that important to mention it? Why the value for DDGS is different of the one indicated in the text (27.40% table 1 vs 27.23% L86).

Response: Values of CP in brackets deleted. The difference in (27.40% table 1 vs 27.23% L86) is just a typing error.

It seems that the diet formulation did not consider digestible AA. It would be better to balance diets on digestible AA rather than on crude proteins. Digestible Amino Acids have to be presented in Table 1 (at least essential AA).

Response: Values of total and digestible amino acids (according to NRC, 19940) are now presented in Table 1.

L90-93: The use of this EEM should be argued (State of the art and hypotheses are lacking in the introduction). The reader does not really understand why this mixture seems interesting to study.

Response: Cleared in the introduction.    

L97 : please indicate the form of the feed (mash, crumble, pellets)

Response: Mash, cleared in the text.

L101: “individually for each replicate” : does it mean individually (per hen) or for each replicate (of 3 hens) ? I understand that you recorded the egg weight per replicate each day (so you have the weight for 3 hens) and then you expressed the result per hen. Is that right ?

Response: Individually means per replicate (of 3 hens) then expressed the result per hen; cleared in the text.

 L140: The Newman-Keuls test is a post hoc comparison test as mentioned. So I guess you performed an ANOVA before the Newman-Keuls test. (This should be clearly stated.) If the ANOVA was used, you should have checked the distribution of variables before. This has to be mentioned in this part because a statistical analysis that is not adapted to the data can question all the results presented.

Response: Firstly, the normal distribution of variables was performed followed by ANOVA and then the post hoc comparison test (Newman-Keuls); cleared in the text.

In addition, you should state if the statistic unit is the replicate (3 hens) or the individual (1 hen).

Response: The statistical unit was the replicate (3 hens); cleared in the text.   

Results

Tables 2, 3, 4, 5 and 6: you only indicated the probabilities for the interaction DDGS x enzyme. Why did you not present all the probabilities?

- DDGS

- EEM

- DDGS x EEM

Response: All tables were provided with all probabilities along with means of the main effects of DDGS and EEM.

Discussion

L 233. I do not understand the “However”. Why do you mention 52 to 53 weeks of age? L80 you indicated the experiment was carried out from 32 to 42 weeks of age.

Response: It was a typing mistake and corrected.

Overall, the discussion lacks of scientific background. It is good to compare the results with literature data but the authors should also make some assumptions on the biological mechanisms that could explain the results/observations and why it agrees or not with previous findings.

Response: More information and explanations were added to the discussion.

Another point that is missing is the high variability in DDGS composition. This point should be addressed and discussed because it affects the generalization of the results.

Response: Done accordingly.   

It is also surprising that in the introduction you indicated that the use of enzymes is beneficial to improve the digestibility of nutrients but no digestibility results is presented.

Response: Actually, we have already performed the digestibility trail but we did not add it to the paper in order to avoid the big size of the manuscript. Anyway, we have added it as a response to the respected reviewer suggestion.

In comparison to existing papers the authors concluded that 12% of DDGS was the maximum level of DDGS to maintain the performances whereas by instance Deniz et al (2013) mentioned 15%. This should be better discussed.

Response: Reasons may be that the study materials were not the same for the two studies such as:

1-       Difference between DDGS used in our study and that of Deniz et al (2013).

Items (%)            

DDGS of   our study

DDGS of   Deniz et al (2013)

Deniz et al. / Livestock Science   (page 176)

Dry matter

89.05

89.48

Crude   protein

27.23

24.94

Crude fat

11.27

11.23

Crude   fiber

7.45

8.47

Crude ash

4.62

4.11

2-       Difference between laying hen strains (Hisex Brown laying hens in our study vs. 480 Super Nick white-laying hens in Deniz et al (2013, page 175).

3-      Age of hens was 31 weeks in our study, however it was 28 weeks in Deniz et al (2013, page 175).